# Strainberry: automated strain separation in low-complexity metagenomes using long reads

Riccardo Vicedomini [1✉], Christopher Quince[2,3,4], Aaron E. Darling [5] & Rayan Chikhi [1]

High-throughput short-read metagenomics has enabled large-scale species-level analysis and functional characterization of microbial communities. Microbiomes often contain multiple strains of the same species, and different strains have been shown to have important differences in their functional roles. Recent advances on long-read based methods enabled accurate assembly of bacterial genomes from complex microbiomes and an as-yet-unrealized opportunity to resolve strains. Here we present Strainberry, a metagenome assembly pipeline that performs strain separation in single-sample low-complexity metagenomes and that relies uniquely on long-read data. We benchmarked Strainberry on mock communities for which it produces strain-resolved assemblies with near-complete reference coverage and 99.9% base accuracy. We also applied Strainberry on real datasets for which it improved assemblies generating 20-118% additional genomic material than conventional metagenome assemblies on individual strain genomes. We show that Strainberry is also able to refine microbial diversity in a complex microbiome, with complete separation of strain genomes. We anticipate this work to be a starting point for further methodological improvements on strain-resolved metagenome assembly in environments of higher complexities.

[1] Sequence Bioinformatics, Department of Computational Biology, Institut Pasteur, Paris, France. [2] Organisms and Ecosystems, Earlham Institute, Norwich, United Kingdom. [3] Gut Microbes and Health, Quadram Institute, Norwich, United Kingdom. [4] Warwick Medical School, University of Warwick, Coventry, United Kingdom. [5] The iThree Institute, University of Technology Sydney, Ultimo, NSW, Australia. ✉email: riccardo.vicedomini@pasteur.fr

The analysis of bacterial communities through metagenome sequencing is a computationally challenging step which is further hampered by strain variation among species. Current techniques for de novo metagenome assembly are able to reconstruct the chromosomal sequences of sufficiently abundant species within a microbial sample, but ideally should also aim at reconstructing each strain present. A critical application of metagenome assembly is the identification of fine differences in the genetic makeup of organisms that end up playing major functional roles in an environment. Sequencing studies of pathogens provide evidence that microbial phenotypes could be strain-specific[1,2]. For example, unusual *Escherichia coli* strains could be highly pathogenic[3] or carcinogenic[4], different *Helicobacter pylori* strains have been associated with different risks for gastric cancer[5], some variants of *Staphylococcus epidermidis* seem to be associated with psoriatic skin[6].

While recognizing the challenges for the precise characterization of what constitutes a "strain", here we will associate the concept of "strain" to a bacterial haplotype, i.e., a contiguous sequence of nucleotides observed jointly and in sufficient abundance by sequencing reads. We therefore define the "strain separation problem" as the reconstruction of partial or complete DNA sequences corresponding to strains, at the base level.

The strain separation problem has been extensively studied in the case of short-read sequencing[7], either without any prior knowledge (de novo) or reference-based. Previous works attempted to tackle the de novo problem using short reads, with various levels of success. DESMAN[8] managed to reconstruct haplotypes of core genes but not entire genomes. STRONG[9] addressed some of DESMAN limitations and is able to determine the number and sequence of strains in a metagenome-assembled genome by exploiting the co-assembly graph over the predicted single-copy-gene sequences and using a variational Bayesian algorithm. ConStrains[10] quantifies and reconstructs *conspecific* (relative to the same species) strain variations by mapping reads on a species-based marker gene set and providing an SNP-based model for each of the identified strains. LSA[11] is a short-read de novo pre-assembly method that aims at separating reads from multiple samples into biologically informed partitions which enable closely-related-strain separation and assembly of individual genomes. OPERA-MS[12] is a hybrid (short and long reads) metagenome de novo assembler, whose ability to assemble genomes at the strain level is dependent on its upstream short-read assembly phase (unassembled sequences and collapsed variants will remain so throughout the pipeline). SAVAGE[13] is a method that performs the assembly of viral quasispecies from deep-coverage short-read sequencing data and is able to reconstruct individual haplotypes of intra-host virus strains. SAVAGE is also employed in VG-flow[14], a de novo approach which enables full-length haplotype reconstruction from pre-assembled contigs of complex mixed samples. All the above methods are not applicable when the input consists of long reads only, and the last two are limited to small genomes (approximately up to 200 Kbp).

Related works have analyzed strains without performing separation, but rather by mapping to existing databases of genomes. StrainPhlAn[15] performs phylogenetic analysis of a population and strain-level variation profiling, through read mapping on reference species-specific markers. StrainEst[16] is a reference-based method exploiting the single-nucleotide variant profiles of available genomes of selected species to determine the number and identity of coexisting strains and their relative abundances in mixed metagenomic samples.

All previous works on strain-level assembly/characterization are based on short-read sequencing data. With the advent of long-read metagenomics sequencing, they cannot be re-used due to fundamental differences in the nature of the data. Many of them (e.g., LSA[11] and references therein) rely on the reliable detection of error-free k-mers with high *k* values (>20), which cannot be performed on long reads due to the high error rate. Some methods[8,10,15] have been specifically designed to work with multiple related samples (e.g., time series), to reconstruct core genes, or to simply provide a strain-level profile of a metagenomic dataset (with no actual assembly). It would be desirable to have a method that, using long reads, produces a strain-separated de novo assembly from only a single sample.

Existing long-read de novo assembly methods can partially, but not completely, separate strains. We will focus on three state-of-the-art assembly methods (metaFlye, Canu, and Lathe), which were either designed for metagenome assembly or have been previously shown to be suitable. metaFlye[17] is, to the best of our knowledge, the only long-read assembler explicitly supporting metagenomic data. metaFlye initially generates a set of error-prone sequences called *disjointigs* representing concatenations of multiple disjoint genomic segments. They are then represented as a repeat graph in which reads are used to untangle repeats and provide an accurate set of contigs. As metaFlye is an option within the Flye assembler, in the following we will refer to metaFlye simply as Flye. Canu[18] is the successor of the Celera Assembler and it was designed for error-prone long reads. It is based on the overlap-layout-consensus paradigm and during its development it introduced new overlapping and assembly algorithms in order to better handle the advances and challenges offered by long-read sequencing technologies. As opposed to Flye, Canu was not specifically designed for metagenomic data although it is often considered an alternative to Flye in this context[19–21]. Lathe[22] is a recently developed workflow for Nanopore reads which attempts to provide complete closed bacterial genomes from microbiomes. It combines existing methods for basecalling, long-read assembly (either Flye or Canu), extensive polishing steps to detect and correct misassemblies, and performs genome circularization.

Throughout the manuscript, we will use the term *strain-oblivious assembly* to refer to a metagenome assembly generated by an assembler which did not attempt any strain separation. Conversely, the term *strain-aware assembly* will be employed to refer to a metagenome assembly where genomic sequences are expected to be reconstructed at the strain level. Moreover, as the assembly of strain sequences from strain-oblivious sequences is a very similar problem to the reconstruction of *haplotypes* in a polyploid genome, we will interchangeably use the terms strain and haplotype in the methodological context.

In this work, we present a method that performs strain separation in low-complexity metagenomes using error-prone long-read technologies. Exploiting state-of-the-art tools for variant calling, haplotype phasing, and genome assembly, we present an automated pipeline called *Strainberry*. It achieves single-sample assembly of strains with higher quality than other state-of-the-art long-read assemblers. Strainberry combines a strain-oblivious assembler with the careful use of a long-read variant calling and haplotyping tool, followed by a component that performs long-read metagenome scaffolding. We extensively tested Strainberry on mock communities as well as uncharacterized real samples and showed that reanalysis of existing long-read metagenomic samples unravels uncharacterized strains.

## Results

### A software pipeline for automated strain separation using long-read sequencing data from low-complexity metagenomes.

The Strainberry pipeline is depicted in Fig. 1 and consists of three main steps: (i) haplotype phasing and read separation, (ii) haplotype assembly, and (iii) strain-aware scaffolding. It requires two

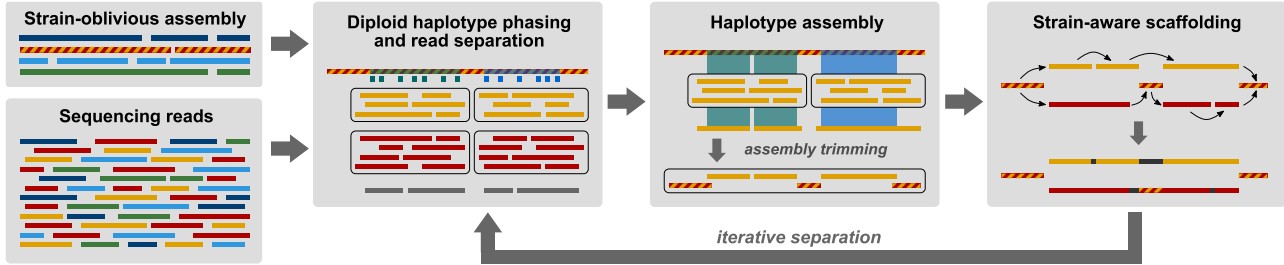

**Fig. 1 Strainberry pipeline.** The pipeline starts from a strain-oblivious assembly and the corresponding set of reads. It then performs haplotype phasing on the strain-oblivious assembly to separate reads into groups that likely correspond to strains. Each group is assembled separately. A final scaffolding step is used to connect sequences likely corresponding to the same strain. The pipeline performs $n-1$ iterations, where $n$ is the maximal number of detected conspecific strains.

inputs: a strain-oblivious metagenome assembly, and a set of long reads aligned to the assembly. The input assembly is either one generated from the input reads, or one made from a combination of sequencing technologies. In our evaluation of Strainberry, we often used a single long-read dataset assembled using the (meta) Flye assembler which is, to the best of our knowledge, the only long-read assembler explicitly supporting metagenomic data. We additionally tested the performance of the Canu assembler. Aligned reads are used to identify single-nucleotide variants (SNVs) on assembled contigs. Sequences containing a large percentage of such SNVs likely correspond to multiple collapsed haplotypes/strains. We use those SNVs to perform haplotype phasing and therefore separate reads originating from different haplotypes. Each separated set of reads can therefore be assembled independently using a standard de novo assembly tool (e.g., Flye). We obtain a strain-aware set of contigs by putting together the non-separated regions of the input assembly, which are either shared between multiple strains or specific to a single strain, and the haplotype-separated assemblies. The last step involves an alignment of input reads on the strain-separated contigs which is exploited to join contigs into longer scaffolds. The three main steps of the pipeline are iterated $n-1$ times, where $n$ is the maximal number of conspecific strains detected by Strainberry. The output of Strainberry is a multi-FASTA file containing contigs shared between multiple strains, and strain-specific contigs. Components of the pipeline are presented in more details in the Methods section.

**Overview of evaluation data.** Datasets of increasing complexity were considered in order to assess to which extent our method is able to reconstruct complete strain genomes. Two low-complexity mock communities (Mock3 and Mock9) containing respectively three and nine precisely-known bacterial strains (Supplementary Table 1) enable an accurate assessment of the quality of results returned by our pipeline. 24 simulated communities were additionally created (or downsampled from Mock3) in order to evaluate Strainberry's ability to assemble conspecific strains characterized by different levels of coverage, divergence, number of strains, and recombination rate. Two metagenomic samples of real sequencing data, one with four dominant bacterial strains and another one with a higher number of species, enabled us to reconstruct novel sequences of strains. They were selected from two different studies. The first one is a low-complexity microbiome from natural whey starter cultures for which two strains of *Lactobacillus helveticus* were identified and assembled[23]. Reads were sequenced with both PacBio and Nanopore technologies which allowed us to assess the performance of our approach with different technologies and validate it with available reference genomes. The second study contains a more complex dataset that is based on Nanopore sequencing data from a human stool

microbiome[22]. Previous analyses hinted at the presence of multiple uncharacterized strains: species-level binning yielded near-complete genome reconstructions characterized by a significant percentage of SNVs with respect to the assembled reads.

**Evaluation metrics for strain-oblivious and strain-separated assemblies.** Prior to separating strains, our method hinges on having an initial high-quality strain-oblivious assembly. We consider three criteria to be important for such an assembly, namely, 1) comprehensive species-level representation, 2) low amount of incorrectly duplicated sequences, and 3) high contiguity. Criterion 1) is a necessary (but insufficient) condition for the complete reconstruction of individual strains. Criterion 2) is not an absolute requirement for any strain separation assembler, but ours relies on similar haplotypes being conservatively collapsed rather than erroneously split. In general, duplicated sequences that are highly similar within an assembly either indicate that some strain-specific regions have been correctly assembled or can be produced by incorrect strain separation. Criterion 3) enables to output more contiguous strain-aware assemblies, due to the fact that strain separation is likely to introduce additional fragmentation over the input assembly.

The evaluation of metagenome assemblies is a complex topic due to the lack of an exhaustive knowledge of the species in a sample, let alone closely related strains. Many standard assembly evaluation metrics need in fact to be interpreted differently in the strain-aware context. We detail them in the "Strain-aware assembly evaluation" section in the Methods. Nevertheless, we will report standard evaluation metrics such as number of contigs, size of the assembly and N50 (i.e., the length for which all contigs of that length or longer cover at least half of the total assembly length) on all the evaluated assemblies.

Specific to synthetic data, the NG50 variant (i.e., the length for which all contigs of that length or longer cover at least half of the total genome length) is reported instead of the N50 value. We will also report the percentage of unaligned bases in both references and assemblies, which is a key metric for evaluating whether complete strain-level genomes were reconstructed. In addition, the average sequence identity, the duplication ratio (i.e., the total number of aligned bases in the assembly divided by the total number of aligned bases in the reference), and the number of misassemblies, are also reported with respect to known reference genomes.

Specific to real datasets, for which a comprehensive set of reference genomes was not available, we used CheckM[24] to quality-control assemblies. CheckM is a reference-free tool that identifies and counts single-copy genes (SCGs) in order to estimate completeness, contamination, and strain heterogeneity of binned contigs. Completeness approximately refers to the percentage of unique SCGs found within a bin, while

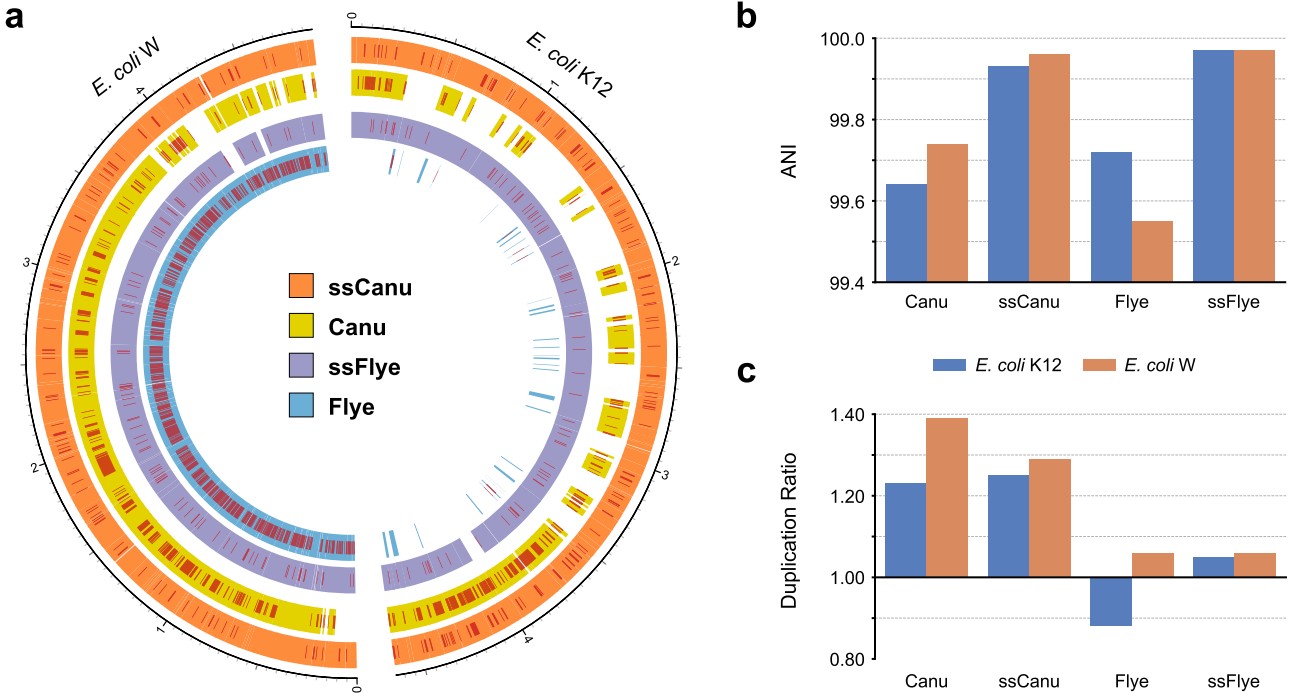

**Fig. 2 Mock3 dataset assembly statistics. a** Circos[58] graph displaying the coverage and SNV-rich regions (single-nucleotide errors) of the strain-oblivious assemblies (Flye and Canu) and their strain-separated counterparts obtained with Strainberry (ssFlye and ssCanu, respectively) compared to the reference sequences of the two *E. coli* strains present. The external graduated scales reflect the genomic positions of the corresponding reference genomes. **b** Average nucleotide identity (ANI) and **c** duplication ratio of assemblies.

contamination reflects the multiplicity of the identified SCGs (as only one copy is expected in each assembled/binned species). Strain heterogeneity is a measure related to the contamination and expresses the proportion (as an index from 0 to 100) of multi-copy marker genes that likely come from closely related species. An important aspect to keep in mind regarding CheckM is that it only works well when genomes/bins are relatively complete. A wrong estimate of completeness and contamination is in fact possible for incomplete bins. We tried to avoid this problem by restricting the analysis of our results on highly contiguous contigs that likely represent complete genomes.

**Separation of two *E. coli* strains in a three-strain mock community**. Mock3 consists of reads sequenced from *Bacillus cereus*, and *Escherichia coli* strains K12 and W. The Flye and Canu assemblers were run on Mock3 and reads were mapped back to each assembly, and we ran Strainberry on both assemblies in order to separate the two *E. coli* strains. Figure 2 shows reference coverage, average nucleotide identity and the duplication ratio of strain-oblivious and the resulting strain-aware assemblies. Strains K12 and W align to each other with an average identity of 98.65% and, for this reason, their entire genomes are collapsed by Flye into two consensus sequences (2 Mbp and 2.8 Mbp long, assembly graph shown in Supplementary Fig. 1). Canu has performed partial strain separation, however it missed the 46% and 11% of *E. coli* strains K-12 and W references respectively, and yielded an unsatisfactory identity percentage (99.64% and 99.74% for strains K-12 and W respectively). The high duplication ratio of the Canu assembly (1.23 and 1.39 for the two *E. coli* strains) excludes it from being characterized as strain-oblivious. Strainberry is able to accurately separate both the Canu and Flye assemblies into two sets of contigs (total length 9.8 Mbp with Flye) with an almost complete coverage of both reference genomes and a nearly perfect average nucleotide identity (with Flye, 99.97% for both *E. coli*

strains). The high duplication ratio inherited from the upstream Canu assembly was slightly reduced.

Overall, the strain-aware assemblies are of high quality, as evidenced by NG50, sequence identity, duplication analysis, and misassembly analysis (Supplementary Note 1 and Supplementary Data 1). As an example, the strain-separated Flye assembly yielded very high NG50 values, duplication ratios close to 1, and a reasonable amount of misassemblies with respect to all references. The Flye assembly had 2 detected inversions and 3 detected relocations, while Strainberry produced 2 detected inversions and 6 detected relocations. We checked that the locations of these putative misassemblies are at the same chromosomal locations in both strains, except in *E. coli* W where Flye generated two short insertions and Strainberry failed to reconstruct two short repetitive sequences (Supplementary Note 1). Nevertheless, all identified misassemblies do not correspond to major rearrangements and, even though the amount of detected misassemblies is numerically higher, this is a natural consequence of having assembled both *E. coli* strains and a significantly larger number of bases.

**Separation of five close strains in a nine-strain mock community**. The more challenging Mock9 dataset is composed of 9 genomes (Supplementary Table 1) with two strains of *E. coli* and two strains of *S. aureus*; the other genomes are of different species. As done in Mock3, a Flye assembly, a Canu assembly, and their respective read alignments were generated and provided as input to Strainberry. In this case, we aim at achieving a two-strain separation of *S. aureus* genomes, and a three-strain separation of *S. sonnei* and the two *E. coli* strains, which are three similar genomes despite *S. sonnei* being a different species. In fact, the two *E. coli* strains, *S. sonnei* but also *K. pneumoniae* all share high sequence identity. Flye was able to separate the *K. pneumoniae* genome, while it collapsed the other genomes into consensus contigs (Fig. 3), as expected from a good strain-oblivious

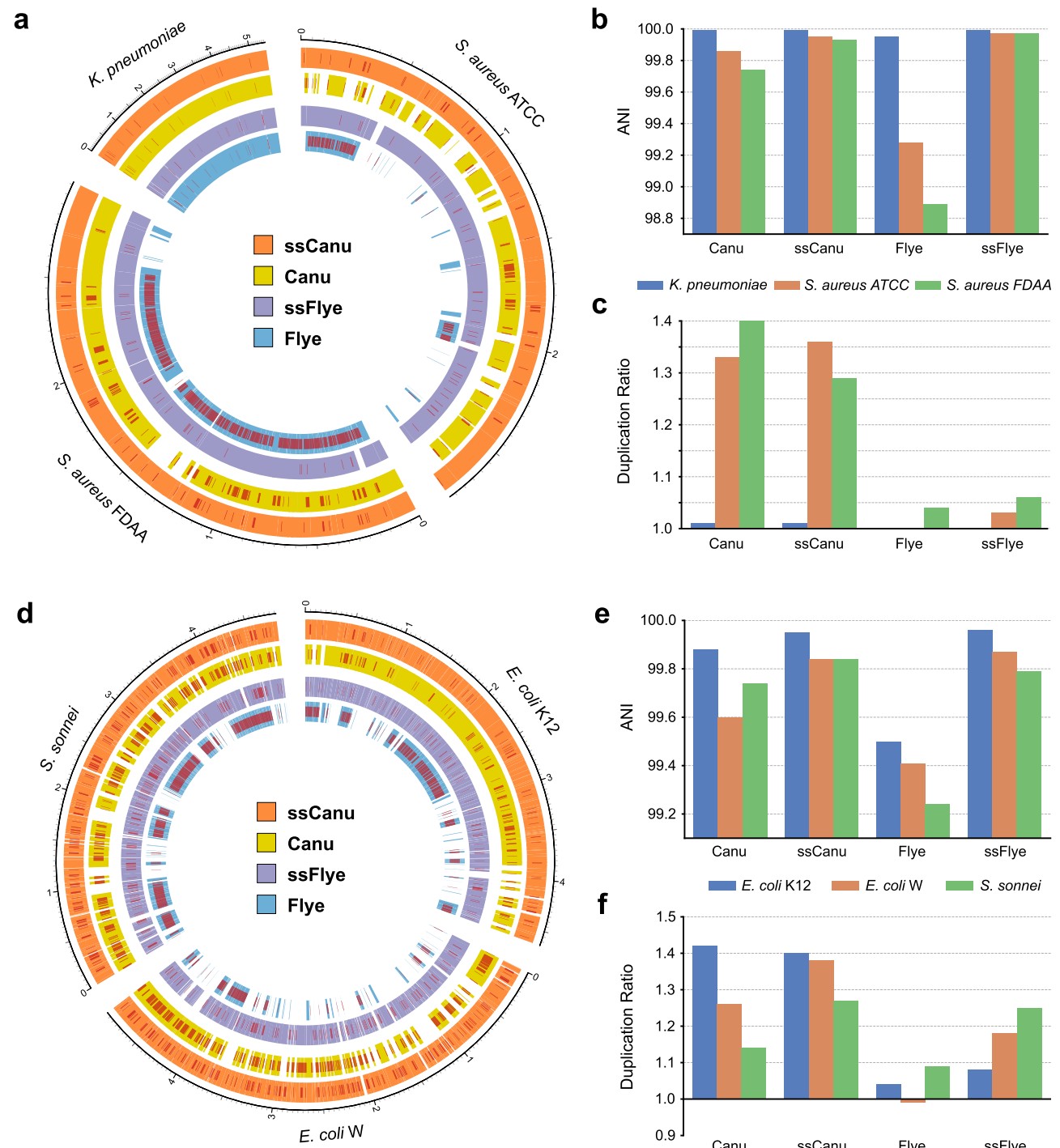

**Fig. 3 Mock9 dataset assembly statistics. a** Circos[58] graph depicting reference coverage and SNV-rich regions of the strain-oblivious assemblies (Flye and Canu) and their strain-separated counterparts obtained with Strainberry (ssFlye and ssCanu, respectively) compared to the reference sequences of *K. pneumoniae* and the two *S. aureus* strains. The external graduated scales represent the genomic positions of the corresponding reference genomes. On the right-hand side, **b** the average nucleotide identity and **c** the duplication ratio of each assembly are reported. The ssFlye and ssCanu assemblies were obtained as a result of a Strainberry separation of the Flye and Canu assemblies, respectively. **d** Circos graph depicting reference coverage and SNV-rich regions of the strain-oblivious assemblies (Flye and Canu) and the strain-separated ones (ssFlye and ssCanu) with respect to the reference sequences of *S. sonnei* and the two *E. coli* strains. On the right-hand side, **e** the average nucleotide identity and **f** the duplication ratio of each assembly are reported.

assembler. Canu, on the other hand, completely reconstructed the *K. pneumoniae* genome but also attempted to separate individual *S. aureus*, *E. coli*, and *S. sonnei* strains. Nevertheless, while providing a significant coverage of the reference sequences, the Canu assembly is characterized by an imperfect sequence identity and a high duplication ratio (Fig. 3).

We first look at the two *S. aureus* strains. They have a lower sequence identity (97.61%) compared to the two *E. coli* strains (98.65%), Flye is still unable to resolve them (Supplementary Fig. 2). Instead, Strainberry correctly separated the Flye consensus sequences into an almost complete coverage of the reference genomes (Fig. 3a), an average nucleotide identity of 99.97% for

both *S. aureus* strains ATCC and FDAA scaffolds (Fig. 3b), and a duplication ratio close to 1.0 (Fig. 3c). Moreover, the final scaffolding step significantly improved contiguity achieving NG50 values of ~2.2 Mbp and ~2.7 Mbp for the two separated strains (Supplementary Data 1). The *K. pneumoniae* genome assembly is also slightly improved by Strainberry which yielded an average nucleotide identity of 99.99% compared to the 99.95% of Flye assembly. The duplication ratio close to 1.0 additionally highlights the good quality of the strain-separated assembly. Canu attempted to separate closely related strains, however with unsatisfactory results. Even though at least 70% of each reference genome was represented in the output assembly, the average nucleotide identity is 99.86% and 99.74% for *S. aureus* strains ATCC and FDAA, respectively. The Strainberry separation of Canu assembly greatly improves these metrics, yielding near-complete assemblies of *S. aureus* strains (Fig. 3a) and 99.9% sequence identity (Fig. 3b). Unfortunately, the high Canu duplication ratio propagates through the Strainberry assembly, even though duplication ratio is slightly reduced after separation and scaffolding (Fig. 3c).

We now turn to the two *E. coli* strains and *S. sonnei*. In this case, Flye is only able to partially separate some strain-specific regions of the three genomes, however with much lower sequence identity compared to the other near-complete assemblies (*B. cereus*, *K. pneumoniae*, *L. monocytogenes*, and *N. meningitidis*). Strainberry significantly improved the reference coverage and the average nucleotide identity with respect to the reference sequences, with duplication ratios of 1.08, 1.18, and 1.25 for *E. coli* strain K-12, strain W, and *S. sonnei* respectively (Fig. 3d–f). Canu was unable to assemble the three reference sequences completely and, at the same time, yielded a very high duplication ratio (as previously observed in *S. aureus* assemblies and in the Mock3 dataset). The Strainberry separation of the Canu assembly overall improved all metrics or yielded comparable results (Supplementary Data 1).

Overall, Flye yielded a better strain-oblivious assembly compared to Canu on Mock9. Strainberry was hence able to benefit from this and provided higher quality assemblies compared to the separation of the Canu assembly, as evidenced by NG50, sequence identity, duplication ratio, and misassembly analysis (Supplementary Note 2 and Supplementary Data 1).

**Influence of strain coverage, divergence, number of strains, and recombination rate on strain separation.** In order to better evaluate Strainberry's ability to assemble strains, we additionally generated 24 mock communities consisting of conspecific strains with different levels of strain coverage, divergence, number of strains, and recombination rates. These communities were assembled with Flye and separated with Strainberry. A comprehensive table of all the assembly evaluation metrics employed in this study is available in Supplementary Data 1.

The influence of strain coverage was tested on mock communities generated by downsampling the Mock3 dataset at 5×, 10×, 20×, 30×, 40×, and 50×. Two downsampling approaches were considered: a uniform depth of coverage of the three strains, and an uneven depth of coverage. The latter consists in applying the downsampling levels to *E. coli* strain W, while keeping the other two strains at 50×. Figure 4 shows, for both the uniformly and unevenly downsampled datasets, that the quality of Strainberry separation reaches a plateau after 30× coverage and is mediocre at 10× coverage. A near-complete reference representation (>95%) and a high sequence identity (>99.8%) thus requires at least 20× coverage. Also, uneven coverage of conspecific strains does not seem to have an impact

on Strainberry's ability to assemble their genomes which is only affected by absolute depth of coverage.

Strainberry's separation of low-divergence strains was evaluated on seven two-strain communities consisting of the *E. coli* strain K-12 and a second *E. coli* strain characterized by an increasing divergence. Datasets were generated with PacBio-like reads that were simulated from known reference genomes (see Supplementary Table 2). In this case, Strainberry fails to separate strains with low strain divergence (likely due to its default behavior of not separating regions with a low percentage of SNVs). A significant improvement of the reference coverage is observed at 0.39% of strain divergence, while a near-complete assembly is reached at 0.50% of divergence. In all these datasets, the average nucleotide identity of separated sequences is always greater than 99.8% and a significant improvement compared to the strain-oblivious Flye assembly (Fig. 4).

In order to evaluate Strainberry with respect to the number of conspecific strains, we created four simulated mock communities consisting 2, 3, 4, and 5 *E. coli* strains characterized by pairwise divergences ranging from 0.7% to 1.4%. Each simulated dataset was assembled with Flye and separated with Strainberry. Figure 4 indicates that Strainberry is able to handle the separation of up to 5 strains albeit with some noticeable loss of quality when separating 5 strains compared to the separation of 3 strains: the average reference coverage drops from 95% (3 strains) to 75% (5 strains), while the average nucleotide identity of sequences with respect to their closest reference drops from 99.9% (3 strains) to 99.6% (5 strains). Nevertheless, in all these scenarios Strainberry improved upon the Flye assembly (Supplementary Data 1).

We also performed an experiment to estimate the influence of historical recombination on strain separation, not including recent recombination events among strains in the same community as they fall outside of our focus on low-complexity communities. Strainberry's ability to discriminate strains depends on the average frequency of SNVs. With higher recombination rates the SNV distribution becomes overdispersed. Nevertheless, recombination tracts tend to be small (a few Kbp, often less) and are expected to be mostly spanned by long reads. In order to evaluate the performance of Strainberry on strains characterized by different historical recombination rates[25], we simulated a mock community based on the complete reference genomes of two *Buchnera aphidicola* strains (low recombination rate), two *Escherichia coli* strains (average recombination rate), two *Helicobacter pylori* strains, and two *Neisseria meningitidis* strains (high recombination rate). As opposed to Flye, which provided strain-oblivious assemblies for the species characterized by average and high recombination rates, Strainberry was always able to separate strain genomes. Nevertheless, the average nucleotide identity of separated assemblies slightly decreases with higher recombination rates (Supplementary Data 1).

**Automated strain separation on a single sample validated by manually processed and heavily curated multi-technology data.** We applied Strainberry to the low-complexity microbiome NWC2 from a natural whey starter culture[23]. NWC2 contains four dominant bacterial species and, among them, two strains of the same species (*L. helveticus* strains NWC_2_3 and NWC_2_4). Therefore, it provides a good test case for our approach. Moreover, four high-quality reference sequences have been produced using a combination of Illumina, Nanopore and PacBio long-read sequencing data, and the following manual curation: Nanopore-based assembly followed by an extensive number of steps involving three PacBio-based, three Illumina-based, and two Nanopore-based runs of polishing. We independently separated

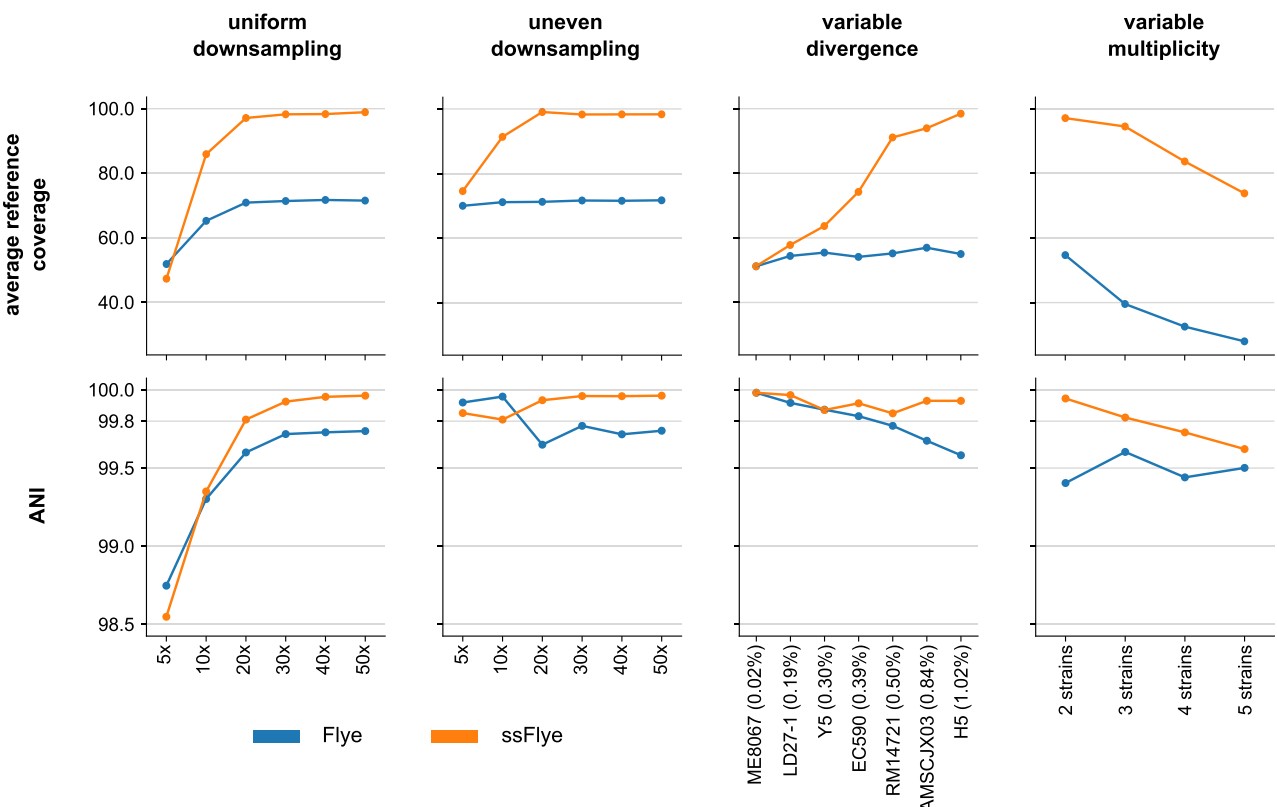

**Fig. 4 Evaluation of strain-separated assemblies with respect to strain coverage, divergence, and number of strains.** Average reference coverages and nucleotide identities of the strain-oblivious Flye assemblies and the Strainberry separated assemblies (ssFlye) on simple mock communities characterized by variable strain coverage (first two columns), divergence (third column), and number of strains (fourth column). The variable coverage communities are downsampled versions of the Mock3 dataset (*B. cereus, E. coli* strain K-12, and *E. coli* strain W). We kept the same depth of coverage among the strains (uniform downsampling) or a constant 50× coverage for *B. cereus* and *E. coli* strain K-12 while downsampling exclusively *E. coli* strain W (uneven downsampling). The variable divergence datasets consist of simulated reads from two strains where one is *E. coli* strain K-12 and the other one is listed on the x-axis (with the divergence percentage shown in parenthesis). The datasets with variable number of strains contain 2, 3, 4, and 5 conspecific strains of *E. coli* with pairwise divergences ranging from 0.7% to 1.4%.

strains using Strainberry applied to either the Nanopore or the PacBio dataset, without ever using short reads, in order to show that we are able to separate strains without the need of multi-technology data nor polishing, and also to assess potential differences between the two long-read technologies. A comprehensive overview of all assembly evaluation metrics computed with respect to the manually curated references are shown in Supplementary Data 2. Significant differences between strain-oblivious and strain-aware assemblies mainly concern *L. helveticus* strain NWC_2_3 and *L. delbrueckii* (Fig. 5), which are discussed next.

*Strain separation using the PacBio technology.* A strain-oblivious assembly of the NWC2 PacBio data was generated with Flye and separated with Strainberry. We also considered an assembly produced with Canu for comparison purposes, even though Canu produced more duplications in the Mock3 and Mock9 experiments.

Figure 5a shows that the original Flye assembly, without any additional polishing, is mostly a strain-oblivious assembly. It is missing 23.03% of the reference of *L. helveticus* strain NWC_2_3 (Fig. 5a) and 19.99% of *L. helveticus* strain NWC_2_4 (Supplementary Data 2). Instead, the scaffolds produced by Strainberry cover almost fully the reference strains (only 9.87% and 0.64% of the strains missing, respectively). Conversely, the original Canu assembly was able to cover almost completely

both *L. helveticus* strains but Strainberry was still able to slightly improve the percentage of unaligned bases of *L. helveticus* strain NWC_2_3 from 7.65% to 5.26%. A significant number of regions of *L. delbrueckii* were also covered twice in all assemblies (Fig. 5b), suggesting the presence of another conspecific strain in the dataset. Overall, Strainberry significantly improved Flye reconstruction of the four dominant strains with 20% additional genomic material for *L. delbrueckii* and *L. helveticus* strains, comparable (or improved) sequence identity, and a reduced number of major misassembly events compared to the Flye/Canu assemblies (see Supplementary Note 3 and Supplementary Data 2).

Additional quality assessment of the assembled strains was carried out using CheckM (Supplementary Data 2). The Strainberry+Flye assembly displays higher completeness values compared to the strain-oblivious Flye assembly for the two conspecific strains (*L. helveticus* strains) while yielding comparable values for other single species (*S. thermophilus* and *L. delbrueckii*). The higher reported contamination in *L. delbrueckii* and *L. helveticus* NWC_2_4 is likely due to the number of duplicated sequences in the upstream Flye assembly, or to the presence of other closely related strains in the metagenomic sample (only the reference sequences of the four dominant strains were available and used for the evaluation). With Canu, no major differences between the strain-oblivious and the strain-separated assemblies were observed.

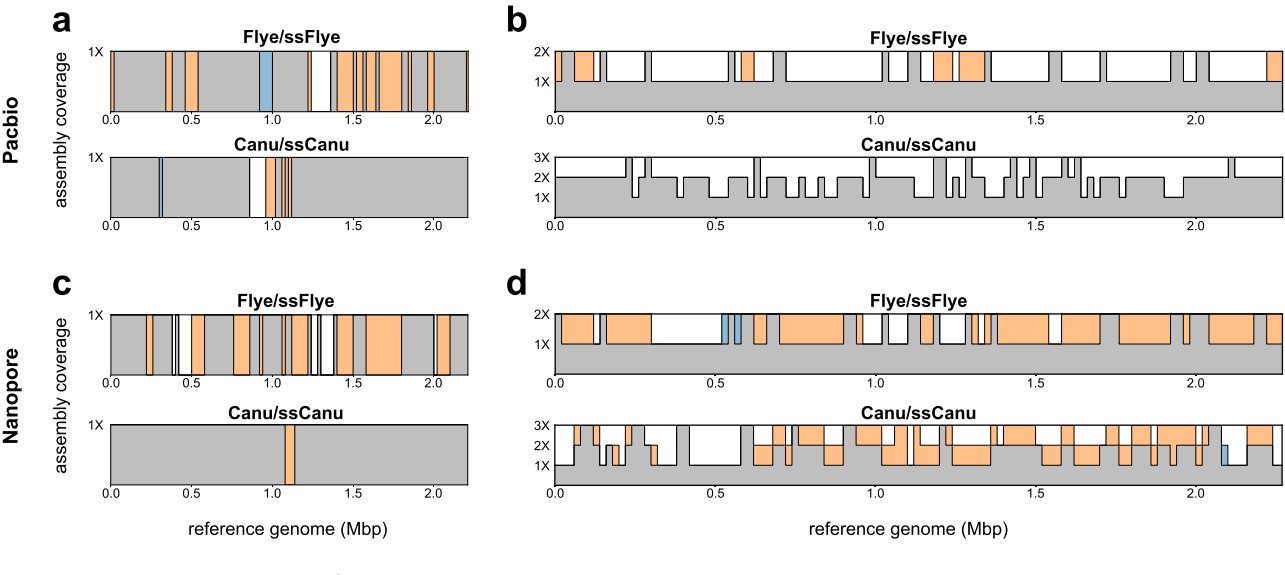

**Fig. 5 Assembly-level coverage of *L. helveticus* NWC_2_3 and *L. delbrueckii* NWC_2_2.** Comparison between the reference coverage of the Flye and Canu assemblies and their strain-separated counterparts generated by Strainberry (ssFlye and ssCanu, respectively) for the following references and datasets: **a** *L. helveticus* NWC_2_3 reference and PacBio dataset; **b** *L. delbrueckii* NWC_2_2 reference and PacBio dataset; **c** *L. helveticus* NWC_2_3 reference and ONT dataset; **d** *L. helveticus* NWC_2_3 reference and ONT dataset. Orange regions highlight a higher coverage of the strain-separated assembly (ssFlye or ssCanu), blue regions highlight a higher coverage of the strain-oblivious assembly (Flye or Canu), and gray regions represent the common coverage level shared by both assemblies.

*Strain separation using the Oxford Nanopore technology.* Similar to the previous section, a strain-oblivious assembly of the NWC2 Nanopore dataset was generated with Flye using a similar protocol[23]. Also in this case a Canu assembly was produced and provided to Strainberry for comparison purposes.

Results follow the same trend as with the PacBio data (see Supplementary Note 4 and Supplementary Data 2). Strain-separated assemblies are able to represent reference sequences almost completely and yield just a slight improvement compared to the PacBio data for *L. helveticus* strain NWC_2_3 (Fig. 5c). There are however some key differences for the assemblies generated with Nanopore data: lower sequence identity to reference strains (98.78–99.76%) than with PacBio data (99.69–99.97%) and a significant increase of strain-separated bases aligning against *L. delbrueckii* reference (Fig. 5d). Polishing of strain-separated contigs using Nanopore-specific tools such as Medaka[26] or MarginPolish[27] would further improve the identity of the *Lactobacillus* strains assemblies (Supplementary Table 3). Overall, thanks to the longer Nanopore reads, Strainberry provided 25–43% additional genomic material on *L. delbrueckii* and *L. helveticus* strains compared to the Flye/Canu assemblies (see Supplementary Note 4 and Supplementary Data 2).

As expected, the CheckM analysis also follows the same trend as with PacBio data. The main difference in the Nanopore dataset is that the putative separation of a conspecific strain of *L. delbrueckii* is even more pronounced, possibly thanks to the longer reads. Moreover, a completeness greater than 90% could be achieved only for some of the strains, regardless of the assembly method. The reason for this behavior is likely due to limitations of Nanopore chemistries so far, yielding more contiguous assemblies at the expense of higher error rates which affect sequence identity and thus completeness (erroneous indels introduced during the assembly of separated reads likely affect marker gene annotation).

**Separation of strains in a human stool microbiome dataset reveals hidden microbial diversity in a curated metagenome assembly.** We applied our pipeline to a realistic strain separation instance, with no prior knowledge about strain variability. A healthy human stool microbiome (patient P1 analyzed in the work of Moss et al.[22]) was sequenced with Nanopore technology. A strain-oblivious reference metagenome assembly was previously generated using the Lathe workflow[22], which included extensive polishing, on which we performed a strain separation using Strainberry. We also created a Flye assembly for comparison purposes. In the following, we will refer to the Nanopore raw data as HSM and to the strain-oblivious reference assembly as Lathe. Compared to NWC2 and the mock datasets, HSM is characterized by a higher species diversity[22]. A MetaBAT2[28] binning of the Lathe reference yields 30 bins of size greater than 2 Mbp, providing a rough estimate of the number of assembled species.

Strainberry displays an increased total length (+19%) and duplication ratio (+0.365) with respect to the Lathe reference assembly (Supplementary Data 3), indicating that additional sequences (strains, as we will see next) were separated. The strain-oblivious Flye assembly also shows a slight increase in duplication ratio (+0.094) showing that it either separated some conspecific strains and/or assembled additional low-abundance material. The latter hypothesis is supported by the presence of unaligned sequences (of total size 34.6 Mbp). Compared to the Flye assembly, the Strainberry assembly also has unaligned sequences, albeit an order of magnitude less than the Flye assembly (total size 2.4 Mbp). While nearly all strain-separated scaffolds align (at least partially) to the reference metagenome, a reduced genome fraction (~90% for Flye vs. ~86% for Strainberry, see Supplementary Data 3) is likely caused by the lower coverage of separated read sets, or, for the Flye assembly, by its different assembly strategy. The number of misassemblies, mismatches and indels shown in Supplementary Data 3 do not accurately reflect

the quality of the assembly as they are obtained by mapping an assembly to the possibly imperfect and strain-oblivious Lathe reference assembly. Precisely evaluating misassemblies would require a perfect metagenome reference which is not available here (see "Evaluation metrics for strain-oblivious and strain-separated assemblies").

Due to the lack of an accurate strain-level characterization of HSM, to assess the ability of our pipeline to identify and separate strains we considered contiguous and complete bins that were likely to represent a consensus of conspecific strains or closely related species. We focused on two nearly complete bins with low contamination (as reported by CheckM), which were likely to contain multiple strains (Supplementary Data 3). They were classified as *Veillonella atypica* and *Eubacterium eligens* at the species level. Note that, as of July 2020, *Eubacterium* is a provisional genus name in the NCBI taxonomy. Bin sizes (2.4 and 3.2 Mbp) were also compatible with the median genome length of the assigned species (2.1 and 3.0 Mbp). In addition, Strainberry produced separations for the *V. atypica* and *E. eligens* contigs as they had a high enough percentage of SNVs (see Supplementary Fig. 3 and Methods).

Depth of coverage and sequence length for *V. atypica* and *E. eligens* scaffolds were evaluated before and after our pipeline as a first evidence of a plausible strain separation. The separated sequences of *V. atypica* are characterized by a halved depth of coverage compared to the Lathe contigs, suggesting two equally abundant strains (Supplementary Data 3). The *E. eligens* contigs also exhibit a similar behavior after the separation (Supplementary Data 3). In both cases, strain separation introduced additional assembly fragmentation (Supplementary Fig. 4).

*V. atypica* and *E. eligens* sequences were further evaluated with CheckM in order to estimate the completeness and the contamination levels of the assemblies (Supplementary Table 4). As opposed to the Lathe reference, both the strain-separated and Flye assemblies yielded a lower completeness. As expected, strain-separated scaffolds also displayed higher contamination and strain heterogeneity. Sequences have indeed been "duplicated" through strain separation and, for this reason, marker genes are found in multiple copies. The lower completeness of strain-aware contigs is likely due to the lower depth of coverage of the assembly (e.g., from 40× to 20× for *V. atypica*) and lack of polishing. Indels appear to be one of the primary causes of frameshifts in open reading frames and hence missing marker genes. A similar behavior was previously observed in the Nanopore-based NWC2 dataset. In order to provide a fair comparison with the Lathe assembly, we polished the Strainberry and the Flye assemblies as carried out in the Lathe workflow (i.e., 4 rounds of Racon[29] followed by one round of Medaka[26]). With additional polishing Strainberry yielded comparable completeness and much higher contamination and strain heterogeneity compared to the Lathe assembly. This is indeed an additional evidence that conspecific strains might have been separated. The polished Flye assembly, on the other hand, was only characterized by a limited improvement in *V. atypica* completeness (Supplementary Table 4). It is worth noticing that the Strainberry separation of the *V. atypica* bin yielded 86% additional genomic material, allowing to characterize separated contigs by two distinct strain-level classifications (~2 Mbp each). Specifically, Strainberry contigs have a best match and moderately cover the *V. atypica* strain ACS-134-V-Col7a and the *V. atypica* strain ACS-049-V-Sch6 available references (Supplementary Fig. 5). Therefore, it appears that HSM contains two conspecific strains of *V. atypica*, one closer to strain ACS-134-V-Col7a, the other one to strain ACS-049-V-Sch6. This result is also observed in the raw reads: 6222 are classified as *V. atypica* at the species level, 2833 (~22.4 Mbp) as strain ACS-134-V-Col7a, and 2504 (~19.4 Mbp) as strain ACS-049-V-Sch6. Strainberry also produced 118% additional genomic material on the *E. eligens* bin, yet it was not possible to classify separated contigs with at a finer level as for *V. atypica*.

We further performed a comprehensive comparison of good-quality Lathe bins (i.e., high completeness and low contamination) with the corresponding strain-separated scaffolds (Supplementary Data 3). More precisely, we defined the bin quality as completeness − 5× contamination (as in ref. [30]) and focused on those bins having a value greater than 70. Strainberry assembly was considered in its polished form. The Kraken2 classification of bin sequences highlighted a finer strain variability after the strain-separation performed by Strainberry (Fig. 6). Several bins also doubled (or increased significantly) in size after the Strainberry separation. Moreover, CheckM reported a completeness greater than 70 and an increase in contamination proportional to the increase in size for seven of the strain-separated bins (Supplementary Data 3).

**Runtimes and memory usage.** All reported analyses have been run on a four-socket computing node with 14-core Intel Xeon Gold 6132 CPUs and 3 TB of memory. For all datasets we reported the time and memory usage of Flye, Minimap2[31], and Strainberry which were run using 12 threads. The first two tools are shown as they were used to generate the input data for Strainberry, except for the HSM dataset in which Strainberry's input consisted in the reference metagenome already available. In all cases, Strainberry required less than 10 GB of memory and less time than Flye according to the dataset complexity and the input coverage (Supplementary Table 5). For instance, in the most complex dataset (the HSM dataset) Flye took 6.5 h and 132 GB of RAM, while Strainberry took 4 h and 9.5 GB of RAM. On the other hand, the Canu assembler always took from 2× to 30× more time than Flye to finish (data not shown).

## Discussion

In this work, we presented Strainberry, an automated pipeline for performing strain-aware metagenome assembly of long reads. Strainberry makes use of well-established tools for haplotype phasing and genome assembly and introduces a strain-aware scaffolding component. Unlike other metagenome assembly methods, it does not rely on either short-read sequencing data or extensive polishing of the assembled sequences. Strainberry is able to accurately separate strains using long reads sequenced from a single metagenomic sample. An average depth of coverage of 60–80× suffices to assemble individual strains of low-complexity metagenomes with almost complete coverage and sequence identity exceeding 99.9%. Strainberry was also able to highlight finer strain variation in a human stool microbial sample which was previously characterized only to the species level. Although Strainberry showed its flexibility by generating strain-aware assemblies regardless of the employed long-read technology, we argue that the use of PacBio sequencing data yields better strain-aware assemblies in terms of sequence identity, while Nanopore sequencing data would favor assembly contiguity. In general, Strainberry can be applied to any long-read dataset of sufficient quality, i.e., for which an initial strain-oblivious assembly can be obtained. In our tests, any strain present at coverage above 10× ends up being correctly separated. The quality of the resulting Strainberry strain separation will also depend on the quality of the initial strain-oblivious assembly.

A key aspect that would merit further study is the relationship between depth of coverage of the sample, read length, and ability to resolve haplotypes. Strains need to be sufficiently covered, and with long enough reads, in order to be fully and precisely resolved. The lack of easy-to-use DNA extraction methods

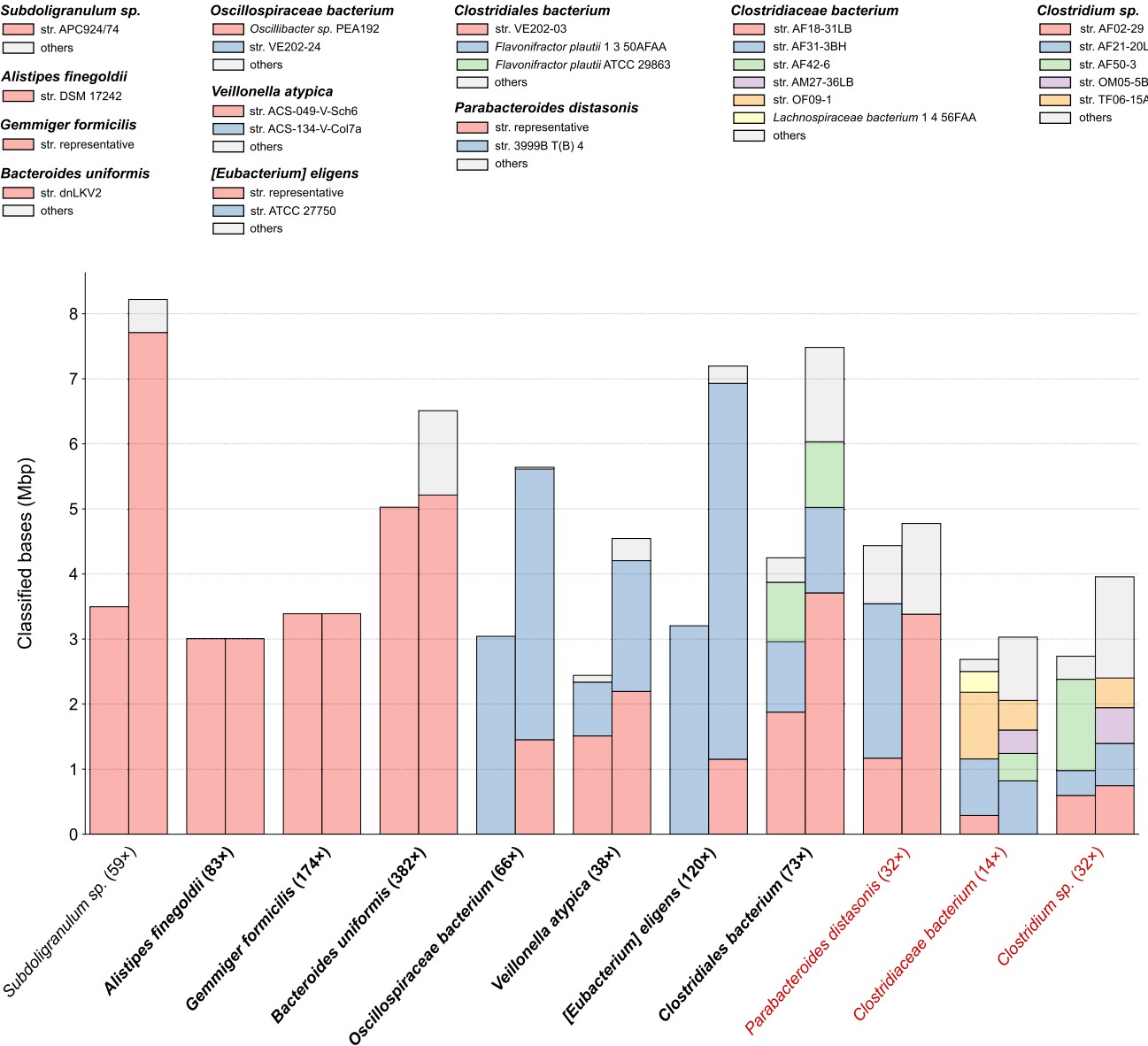

**Fig. 6 Assembly size and sequence classification before and after strain separation.** Bins (x-axis) are named at the species level according to the most dominant Kraken2 classification. The value between parentheses represents the average depth of coverage of the bin before the strain separation. Bins highlighted with a bold font have a moderate post-separation completeness (>70%), while those highlighted in red have either poor completeness (<50%) or low read coverage (<30×). Colored bars represent the number of bases classified as a specific species/strain in a bin before and after the strain separation (left and right-hand bars, respectively). Classified sequences whose size accounts for less than 10% of the bin size have been grouped as "others".

producing high molecular weight DNA for all bacteria in a microbial community limits read length (to around a N50 of 5 Kbp). As we showed on the human stool microbial sample, low-coverage strains are likely separated into shorter contigs/scaffolds compared to the input sequences due to poorly covered regions. This limitation also affects strain-oblivious assemblies, to a lesser extent given that species-level assembly is achieved by pooling the depth of coverage of all strains present.

Strainberry finds strain variations and separates reads by exploiting state-of-the-art haplotype phasing techniques that were designed for genomes. Notably, two recent pipelines that also rely on those techniques produced chromosome-scale haplotype-resolved single human assemblies[32,33] using additional sequencing data (Hi-C or Strand-seq). Haplotype phasing in single genomes benefits from read coverage being evenly split among the different haplotypes. This assumption no longer holds in a

metagenomic setting where haplotypes/strains typically exhibit different abundance/coverage levels. While we showed this is not a limitation for the actual separation of a limited number of well-represented strains, we believe it is an important development worth pursuing. Moreover, in the presence of high strain variability, strain-oblivious sequences might need to be separated into a variable number of strains within the same contig. Finally, we did not evaluate Strainberry's ability to resolve plasmid genomes, which we leave for future work.

## Methods

**Datasets and preparation**. Mock communities were constructed from a 10-plex PacBio Sequel dataset (see "Data availability"). Sequenced reads were demultiplexed with PacBio's tool lima[34] v1.11.0 and assigned to two mock communities: Mock3 and Mock9. The former contains a ~400X coverage of *Bacillus cereus*, *Escherichia coli* strains K12 and W. The latter instead contains a ~430× coverage of the same genomes in Mock3, with the addition of *Klebsiella pneumoniae*, *Listeria*

*monocytogenes*, *Neisseria meningitidis*, *Staphylococcus aureus* strains ATCC 25923 and FDAARGOS 766, and *Shigella sonnei*. The precise depth of coverage of each single genome is available in Supplementary Table 1. Assemblies of Mock3 and Mock9 datasets were generated with Flye v2.7 with parameters --meta (corresponding to metaFlye) and --pacbio-raw. The estimated genome size (--genome-size option) was set to 15 Mbp and 42 Mbp for Mock3 and Mock9, respectively. Reads were aligned to the assemblies using minimap[31] v2.17 with parameter -axmap-pb, sorted with the sort command of samtools[35] v1.10. Due to the very high depth of coverage, the Mock3 and Mock9 alignments were downsampled to obtain approximately 75× and 86× coverage respectively (command samtools view -s 0.15) in order to run our pipeline in reasonable time. For evaluation, reference sequences of the genomes in the mock datasets were downloaded from NCBI (Supplementary Table 1). Assembly graphs for the Flye assemblies (Supplementary Figs. 1 and 2) were generated using Bandage[36] v0.8.1.

The mock communities considered to evaluate Strainberry with respect to strain coverage, divergence, number of strains, and recombination rate were generated as follows. To analyze the impact of strain coverage we downsampled the Mock3 dataset uniformly (all the strains) or unevenly (exclusively *E. coli* strain W), considering the following depths of coverage: 5×, 10×, 20×, 30×, 40×, and 50×. When the only *E. coli* strain W was downsampled, the other two strains were kept at a constant 50× depth of coverage. Strainberry performance analysis with respect to strain divergence was carried out generating 7 two-strain mock communities consisting of simulated reads from the complete genomes of *E. coli* strain K-12 and one of the following *E. coli* strains: ME8067, LD27-1, Y5, EC590, RM14721, AMSCJX03, and H5. The four simulated datasets based on the increasing number of conspecific strains consisted of reads simulated from the first *n* complete genomes (for *n* = 2, 3, 4, 5) of the following *E. coli* strains: K-12, W, H5, LD39-1, and AMSCJX03. Finally, the mock community considered to evaluate Strainberry with strains characterized by different historical recombination rates was simulated from the complete genomes of the following strains: *Buchnera aphidicola* strains Bp and W106, *Escherichia coli* strains K-12 and W, *Helicobacter pylori* strains ASHA-004 and ASHA-005, and *Neisseria meningitidis* strains 09-292 and 11-7. Accession codes of the reference genomes considered for the simulated mock communities are listed in Supplementary Table 2. Reads were generated with Badread[37] v0.2.0 and parameters --quantity 50x --error_model pacbio2016 --qscore_model pacbio2016 --identity 90,97,3 --length 6100,3700 in order to mimic the sequence length and identity of the reads considered in Mock3. Assemblies were generated with Flye v2.7 with parameters --meta and --pacbio-raw. The estimated genome size (--genome-size option) was set to the total length of the known reference sequences of the downsampled/simulated mock community.

The PacBio and Nanopore reads of the NWC2 sample and the reference sequences were retrieved from NCBI (see "Data availability"). The PacBio dataset contains a total of 763,335 reads sequenced using PacBio's Sequel platform, however only the 385,106 reads longer than 5 Kbp were retained for generating an assembly. The Nanopore dataset contains a total of 407,027 reads sequenced with the MinION platform. In this case only the 33,364 reads longer than 10 Kbp were retained for generating an assembly. Both assemblies were constructed with Flye v2.7 using a minimum read overlap of 3 Kbp, an estimated metagenome size of 8 Mbp, and four polishing iterations (options --meta --min-overlap 3000 --genome-size 8 m --iterations 4). Flye was run with parameters --pacbio-raw and --nano-raw for PacBio and Nanopore data, respectively. Reads were aligned to the assemblies using minimap v2.17 with parameters -axmap-pb and -axmap-ont for the PacBio and Nanopore datasets respectively, and position-sorted with the sort command of samtools v1.10. The alignment of PacBio reads was further downsampled to obtain approximately a 66× coverage (command samtools view -s 0.1743). The alignment of Nanopore reads already yielded a 63× coverage of the assembly and therefore no downsampling was applied.

Finally, the HSM dataset was also retrieved from the NCBI database (see "Data availability" for accession numbers). More precisely, 4,003,772 Nanopore reads and the 163-Mbp metagenome assembly generated with the Lathe workflow were downloaded and used as input for Strainberry. Reads were aligned to the Lathe assembly using minimap v2.17 with parameter -axmap-ont and sorted with the sort command of samtools v1.10. Since the alignment yielded a 61× coverage of the assembly, no downsampling was applied. The Flye assembly for this dataset was generated with parameters --meta, --nano-raw, and an estimated metagenome size (--genome-size option) set to 160 Mbp.

All sequencing data available from the NCBI Sequence Record Archive (SRA) was retrieved using the SRA Toolkit[38] v2.10.8. Reference sequences were retrieved using the NCBI's tool Entrez Direct[39] v13.9. All main results (assemblies, evaluation metrics, and plots) presented throughout the manuscript were generated using a custom analysis workflow which is open source and publicly available at the following repository: https://github.com/rvicedomini/strainberry-analyses. The following Python packages were used either in Strainberry or in the analysis workflow: pysam[40], biopython[41], numpy[42], scipy[43], pandas[44], seaborn[45], matplotlib[46], networkx[47], pygraphviz[48], and pyvcf[49].

### Components of the strain separation pipeline

*Haplotype phasing and read separation.* The input of the first step of the strain separation pipeline consists in a strain-oblivious assembly and an alignment of the long reads against it. The rationale is to identify single-nucleotide variants (SNVs) on assembled contigs and use them to separate reads into different haplotypes (each haplotype will then correspond to a strain). Unfortunately, there is a scarcity of variant callers designed to handle efficiently long, error-prone reads, and at the same time supporting ploidy above 2. For this reason, we devised an approach that iteratively performs a diploid separation using the Longshot[50] diploid long-read SNV caller, that is also able to tag (separate) reads according to their most likely haplotype. Read separation is automatically performed by its companion phasing tool HapCUT2[51]. The output of the first step of each strain separation iteration is therefore a set of phased haplotypes (or phasesets) that partition the input assembly and a set of reads for each haplotype.

All variant calling and read separation tools were run with their default parameters. We also considered the use of a combination of freebayes[52] and WhatsHap Polyphase[53] for a non-iterative polyploid separation of the strains. This alternative strategy did not provide major improvements on the mock datasets (Supplementary Data 4) and were not further evaluated. Computational time was also considered to select an appropriate combination of tools (Supplementary Table 6).

*Assembly of strain-aware contigs from separated reads.* A phaseset is usually defined as a list of phased SNVs along a sequence (e.g., a contig), that is SNVs assigned to a specific haplotype. In this work, we slightly redefine a phaseset to be the interval which includes the phased SNVs (and not the SNVs themselves). More formally, given a strain-oblivious contig $C_i$, we define a *phaseset* over $C_i$ as a tuple $PS = (i, s, e)$, where $i$ is the contig identifier, and $s$ and $e$ are the first and last positions, respectively, of the SNVs belonging to PS. Moreover, we define dens(PS) as the percentage of SNVs that have been detected in the interval $[s, e]$ of $C_i$.

*Assembly of haplotype-separated reads.* The separated reads can be independently assembled by a standard (genome) assembly tool. We expect this task to be much simpler compared to the assembly of the whole metagenome, as the read separation step should in principle yield reads that correspond to a single strain. For this reason, and to handle a potentially large number of separated regions, we opted for the fast long-read assembler wtdbg2[54]. Moreover, as phasesets could be identified in quite short regions (couple of thousands bases) and the depth of coverage is reduced by the read separation, we decided to run wtdbg2 with fine-tuned parameters. We slightly increased the minimum read depth of a valid edge to 5 (option -e 5), we decreased the minimum length of alignment to 1000 bp (option -l 1000), we decreased to 3000 bp the threshold to drop short reads (option -L 3000), we disabled k-mer subsampling (option -S 1), and enabled the realignment mode (option -R). Note that these parameters were not tuned according to a particular dataset but were set rationally and identically for all evaluated datasets. Finally, phasesets PS where dens(PS) <0.1% were discarded. This allows, first, to avoid the separation of strains having high sequence identity (greater than 99.9%) and, second, to evade splitting contigs due to phasesets identified due to a small number of false-positive SNVs.

*Assembly post-processing.* The assembly of separated reads posed some additional difficulties that needed to be handled in order to provide an accurate strain-aware assembly. More precisely, let $PS = (i, s, e)$ be a phaseset over the contig $C_i$ defined on the interval $[s, e]$. The depth of coverage of a separated set of reads with respect to PS is expected to progressively decrease at positions $j<s$ and $j>e$, resulting in a low-quality assembly at the extremities of phasesets. Therefore, to reduce this effect, each haplotype-separated assembled contig is mapped to its corresponding strain-oblivious contig $C_i$ and its extremities are trimmed in order to keep exclusively the aligned sequence corresponding to the interval $[s, e]$ of $C_i$. Moreover, the strain-oblivious unphased (non-separated) subsequences longer than 500 bp are retained. The set of the haplotype-assembled and unphased sequences constitutes the Strainberry strain-aware set of contigs.

### Strain-aware scaffolding.

We explored the possibility of implementing a simple scaffolding algorithm to produce a more contiguous assembly. Ideally, strain-aware contigs belong to three types: *strain-resolved* (contigs that align to multiple strains but map better to one than the other(s)), *strain-specific* (contigs belonging to one strain only), *core* (contigs that correspond to regions that are near-exactly shared by both strains). Therefore, aligning the original set of reads against the strain-aware assembly might allow for properly joining strain-resolved contigs, possibly including strain-specific and core contigs in between. Strain-aware scaffolding is implemented as follows. First, the input reads are aligned against the strain-aware contigs. Second, a *scaffolding graph* is built according to those reads whose alignments bridge different contigs. Third, the graph is simplified, and unambiguous paths are processed to generate a set of strain-aware scaffolds.

*Definitions.* A *dovetail alignment* between a contig *C* and a read *r* is an alignment between either the suffix of *C* and the prefix of *r* or vice versa, the prefix of *C* and the suffix of *r*. We define a *(strain-aware) scaffolding graph* based on the string graph formalism[55]. A vertex is associated to each contig (henceforth the two terms will be used interchangeably), while edges are the dovetail alignments, indicating adjacency between contigs. More formally, a *scaffolding graph* is a bi-directed weighted graph $G = (V, E, w)$. $V$ is the set of all strain-aware contigs $S_i$. A bi-

directed edge $e_{ij} \in E$ is put between two vertices $S_i$ and $S_j$ if and only if there exists at least one read $r$ having two dovetail alignments, one with $S_i$ and the other with $S_j$, and those dovetails alignments are such that one must involve the prefix the read, and the other the suffix of the read. Moreover, $e_{ij}$ is modeled as a bi-directed edge by adding oriented arrows at both its endpoints. The direction of the arrow on $S_i$'s side of $e_{ij}$ is defined as follows: if the dovetail alignment involves the suffix of $S_i$, it points away from $S_i$; otherwise, if the dovetail alignment involves the prefix of $S_i$, it points towards $S_i$. The number of reads connecting $S_i$ and $S_j$ defines the weight $w(e_{ij})$ of the edge $e_{ij}$.

During a graph traversal, entering a vertex corresponding to $S_i$ with an arrow pointing towards it, tells us to consider the contig $S_i$ as it is. Otherwise, the reverse complement of $S_i$ must be considered (arrow pointing away from $S_i$). The same reasoning applies to $S_j$. Moreover, when entering a vertex from an arrow pointing towards it, an edge with the arrow pointing away from it must be used to exit, and vice versa.

*Read alignment and filtering.* Reads are mapped against the strain-aware contigs using minimap2[31] with option -cxmap-pb (-cxmap-ont) in order to perform a base-level alignment of PacBio (Nanopore) reads. Unique alignments (i.e., primary minimap2 alignments with mapping quality greater than 40) which are also dovetail alignments are retained in order to build the scaffolding graph. An overhang (i.e., the unaligned extremity) that is the minimum between 50 bp and the 10% of the match length is however allowed for identifying prefix and suffix matches.

*Graph construction and simplification.* The identified dovetail alignments are used to build the scaffolding graph according to the previous definition. In this process, we consider the input strain-oblivious assembly as a "backbone" and we allow only certain links according to the position of contigs on the input assembly. We say that two contigs are *consecutive* if they map to the same backbone contig and no other contig maps in between them. Two haplotype-assembled contigs are *adjacent* if they map to the same backbone contig and no other phaseset exists in between them (i.e., two adjacent contigs are not necessarily consecutive). Two contigs are *read-linked* if there exists at least one read showing evidence of these two contigs being close in a strain. An edge is then allowed only between contigs of the following types: (i) two consecutive contigs; (ii) two adjacent haplotype-assembled contigs; (iii) a non-separated backbone contig and a haplotype-assembled contig which are read-linked; (iv) two contigs both mapping on the extremity of a backbone contig.

Once the scaffolding graph has been constructed it is trimmed to remove simple transitive edges and weak edges. The former is an edge $e_{ij}$ between two vertices $S_i$ and $S_j$ (where $i \neq j$) for which there exists a path $S_i \rightarrow S_z \rightarrow S_j$ from $S_i$ to $S_j$ traversing a distinct node $S_z$ ($z \neq i$ and $z \neq j$) and whose arrows at the sides of $S_i$ and $S_j$ are of the same type as $e_{ij}$. A weak edge is an edge exiting a vertex $S_i$ either being supported by less than 10 reads or such that its supporting reads are less than the 90% of the reads supporting all edges exiting $S_i$. The simplified graph is traversed, and a scaffold is produced for each maximal unambiguous path (i.e., a path without bifurcations that cannot be extended) introducing gaps when needed. More precisely, gap lengths are computed as the median distance between two contigs according to the reads supporting the corresponding edge, when such a distance is positive. Conversely, contigs are simply joined one after the other if the median distance has a negative value.

**Iterative separation.** The main pipeline (haplotype phasing, read separation, haplotype assembly, and scaffolding) is performed in an iterative fashion, as long as the output assembly is likely to be improved by an additional separation. This section describes the process in more detail. Input reads are mapped against Strainberry scaffolds and a tentative haplotype phasing is carried out and "compared" with respect to the previous haplotype phasing. The comparison uses the concept of Hamming rate, defined between a read and a phased haplotype as the ratio between the number of different nucleotides among the shared SNV positions and the total number of shared SNV positions (it is hence a value between 0 and 1). An additional iteration is performed only if the average Hamming rate between reads and their closest haplotype improves.

More in detail, the Hamming rate is computed only for reads that overlap a phased region by at least 3 Kbp and only the smallest value among the phased haplotypes is taken into account (i.e., the ratio corresponding to the closest haplotype). If the average Hamming rate of these reads improves (i.e., is smaller) by at least 1% of the value computed during the previous iteration, the haplotype assembly and the scaffolding steps are subsequently performed. Otherwise, the last strain-separated assembly is returned and Strainberry execution is terminated. In order to prevent unnecessary separations that could worsen the quality and the contiguity of the output assembly, each additional iteration is performed sequence-wise. A new iteration is therefore performed only if the average Hamming rate improves *globally* (i.e., taking into account all phased regions), while the actual haplotype assembly and scaffolding steps are performed exclusively on sequences for which the average Hamming rate improves *locally* (i.e., taking into account only the phased regions of a sequence).

**Strain-aware assembly evaluation.** When only a single reference genome is available for a given species and used to evaluate the assembly of multiple strains, metrics such as sequence identity, duplication rate, and number of misassemblies need to be evaluated with caution. As an example, in an isolate-genome assembly, a duplication ratio much greater than 1 would typically indicate a problem in the assembly. On the contrary, in a strain-aware meta-genome context with limited knowledge of the organisms, a higher duplication ratio might be simply an indication that multiple strains have been resolved. A similar observation could be made when observing a lower sequence identity and/or the presence of misassemblies. The former is expected when only a single reference is compared against strain-resolved sequences as the consensus sequence will diverge from individual strains. On the other hand, misassemblies such as SNVs and indels could be due to strain differences. For these reasons, metrics need to be evaluated differently with respect to the datasets considered in this study. Mock datasets are precisely evaluated as all the genomes are known and accurate references are available. Evaluation metrics computed on the real datasets, on the other hand, need to be analyzed with care.

All generated strain-oblivious and strain-aware assemblies were compared to a set of reference sequences corresponding to genomes and strains known to be present in the sample, when available. Each contig/scaffold was assigned to the best-identity reference sequence using the MUMmer4[56] package. More precisely, a custom script was used to assign each contig/scaffold to its closest reference sequence for which the contig/scaffold aligns for at least 50% of its length. The closest reference sequence is defined as the one for which the score alignedBases × averageIdentity is the highest, where alignedBases is the number of aligned nucleotides of an assembled sequence against a reference and averageIdentity is the average nucleotide identity of the aligned sequence. Each set of sequences assigned to a reference was then compared using the MUMmer4 command dnadiff and the output files were processed to obtain the GAGE[57] assembly evaluation metrics. Circular and bar graphs in Figs. 2 and 3 were generated from the MUMmer output with Circos[58] and custom Python scripts (see "Code availability").

The following metrics were computed: number of contigs, assembly size, N50/NG50, reference and assembly aligned percentages, average sequence identity, number of duplicated/compressed bases, and number of misassemblies. Misassemblies can be divided into four major categories, following the terminology and criteria of QUAST[59]: SNPs, relocations, translocations, and inversions. SNPs are simply single-base mismatches with respect to the reference sequence. Relocations arise instead when the consistent ordering of aligned blocks is disrupted. Translocations are relocations involving two adjacent blocks in the assembly that align to different sequences/chromosomes in the reference genome. Inversions are characterized by subsequences that align on opposite strands.

Assembly-level coverage plots of NWC2 available references (Fig. 5) were produced by aligning the contigs assigned to a reference with minimap2 (parameters -ax asm20 --cs) and computing the assembly-level coverage in 20-Kbp windows with mosdepth[60] (parameters --by 20000 -m -x). CheckM v1.1.2 was run with the Bacteria marker-gene set in order to evaluate completeness and contamination percentages of the assembled sequences.

Some specific and more appropriate evaluation procedures were considered for the HSM real dataset due to the lack of an accurate strain-level characterization and its higher complexity compared to the Mock3, Mock9, and NWC2 datasets. They are precisely detailed in the Supplementary Note 5.

**Reporting summary.** Further information on research design is available in the Nature Research Reporting Summary linked to this article.

## Data availability

All described datasets are publicly available. Links and accession codes are provided in this section. PacBio sequencing data used to create the Mock3 and Mock9 datasets is available at https://github.com/PacificBiosciences/DevNet/wiki/Microbial-Multiplexing:-PacBio-Sequel-System,-Chemistry---v3.0,-Analysis---SMRT-Link-v6.0.0. NCBI RefSeq accession codes of the reference genomes used for the assembly evaluation and the generation of the simulated mock communities are listed in the Supplementary Information (Supplementary Tables 1 and 2). NWC2 reads are available at the NCBI BioSample SAMN09580370 under the SRA accession codes SRX4451758 (Nanopore) and SRX4451757 (PacBio). Reference sequences used for the assembly evaluation of NWC2 are accessible under the NCBI BioSample accession codes SAMN09476686 (*S. thermophilus*), SAMN09476687 (*L. delbrueckii*), SAMN09476688 (*L. helveticus* strain NWC_2_3), and SAMN09476689 (*L. helveticus* strain NWC_2_4). HSM reads and the corresponding metagenome assembly reference are available at the NCBI BioProject PRJNA508395 under the SRA accession code SRX5235113 and the GenBank accession code GCA_011075405.1, respectively. All assemblies generated with metaFlye, Canu, and Strainberry are available at https://doi.org/10.5281/zenodo.4721347.

## Code availability

Strainberry has been developed with the Python programming language and it has been tested on a Linux platform. Strainberry is open source and available at https://github.com/rvicedomini/strainberry under the MIT License. Scripts necessary to reproduce the

main results are available at https://github.com/rvicedomini/strainberry-analyses. Results were generated using the version 1.1 of the analysis workflow and the version 1.1 of Strainberry[61].

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

## Acknowledgements
R.V. and R.C. are funded by ANR INCEPTION (PIA/ANR-16-CONV-0005) and R.C. by ANR PRAIRIE (ANR-19-P3IA-0001) and ANR SeqDigger (ANR-19-CE45-0008). C.Q. is funded through "Strain resolved metagenomics for medical microbiology" MRC MR/S037195/1 and the "CLIMB-BIG-DATA" consortium MR/T030062/1. A.E.D.'s contributions to this research were supported in part by the Australian Government through the Australian Research Council Discovery Projects funding scheme (project DP180101506). The authors thank Eduardo Rocha for his advice on the manuscript.

## Author contributions
R.V. and R.C. conceived the study. R.V. developed the software, carried out all the analyses, and generated the figures. R.V., C.Q., A.E.D., and R.C. wrote the manuscript. All authors read and approved the final manuscript.

## Competing interests
A.E.D. holds equity in and is cofounder and CSO of Longas Technologies Pty Ltd, which is a commercial entity developing synthetic long-read sequencing technologies. The authors declare no additional competing interest.
