## [Peer Review File · Nature Communications]

REVIEWER COMMENTS

Reviewer #1 (Remarks to the Author):

The authors present a software pipeline for the assembly of genome sequences from metagenomic sequences, which represent mixtures of closely related strains. As many microbial species occur as multiple strains in their natural environments, strain separation is a major problem in metagenomic assembly and binning. The manuscript discusses existing, mainly short-read techniques and introduces a workflow that is based on long-read assembly and haplotyping. The approach is limited to low complexity metagenomes and the software requires previous knowledge about the number of strains. Therefore, Strainberry seems to be an intermediate step on the long way towards fully strain-resolved metagenomics.

The manuscript provides very interesting data, which justify most of the conclusions. However, it leaves a couple of major questions open:

1. How different can strains be, to be assembled by Strainberry? The mock community experiments don't answer this question sufficiently.
2. How rare can strains be, to be assembled by Strainberry? Many metagenomes consist of one or few abundant strains and many rare strains.
3. How much does the assembly result depend on the mode of evolution between strains (e.g., amount of recombination compared to inversion and translocation)?
4. What criteria apply to long-read sequencing data to be used by Strainberry? Many datasets are of poor quality - can they be used or are there minimal criteria e.g. for coverage and sequence fractions that can be aligned to the initial assembly? Or should the user just try?

Minor issue:

1. Supplementary Figure 3 lacks a unit for SNP density.

Technical issue:

We failed to install strainberry. All prerequisites were installed.

In the environment.yml, the line that doesn't work is the following:

```
- pip:  
- git+https://github.com/whatshap/whatshap.git@polyploid-haplotag
```

When we replace that line with:

```
-whatshap
```

It does install as I mentioned, but then the multi-strain separation doesn't work.

The error during the installation is the following:

```
ERROR: Failed building wheel for whatshap
```

```
ERROR: Could not build wheels for whatshap which use PEP 517 and cannot be installed directly
```

Here is the part of the error message which makes me think that the problem is in the code and not in our system or the installation procedure:

In file included from src/polyphase/trianglesparsematrix.cpp:1:

```
src/polyphase/trianglesparsematrix.h:15:5: error: 'uint64_t' does not name a type
```

```
15 | uint64_t entryToIndex(uint32_t i, uint32_t j);
```

```
| ^~~~~~
```

```
src/polyphase/trianglesparsematrix.h:16:5: error: 'uint64_t' does not name a type
```

```
16 | uint64_t size();
```

```
| ^~~~~~
```

```
src/polyphase/trianglesparsematrix.h:17:15: error: 'uint32_t' has not been declared
```

```
17 | float get(uint32_t i, uint32_t j);
```

```
| ^~~~~~
```

```
src/polyphase/trianglesparsematrix.h:17:27: error: 'uint32_t' has not been declared
```

```
17 | float get(uint32_t i, uint32_t j);
```

And there are many more instances of similar errors.

Reviewer #2 (Remarks to the Author):

In this paper, the authors propose a new method called Strainberry that uses long reads to identify strains from metagenomic samples and evaluate the accuracy of Strainberry on both synthetic and real metagenomic datasets. Strain resolution is a very important problem in metagenomics and the authors provide good insights into how this problem can be solved using emerging use of long read technologies in metagenomics. The manuscript is very well written and easy to follow. The code for the method is provided on GitHub with detailed instructions to install and run. Overall, the manuscript contributes significantly to the problem of strain resolution.

Here're some of the comments that need to be addressed before I can recommend the manuscript for publication.

- Strainberry method is very much similar to one of the recent methods published for the haplotype assembly of Human genome - <https://www.nature.com/articles/s41587-020-0711-0> . I would recommend the authors to cite this paper and mention that similar approaches have been used for single genome assembly.

- Strainberry requires the number of estimated strains in the sample as an input from the user. Most of the evaluation presented in the paper was based on the samples with the known number of strains for a particular species. However, this seldom happens in real metagenomic samples. the authors allude to this point in the discussion, but some evaluation is required on this. For example, if one runs Strainberry with different values for the number of strains, how they can determine what's the right number of strains.

- The authors use bacterial genome assemblies generated using long reads as a starting point for Strainberry. However, these assemblies tend to have low base level accuracy as the reads are not as accurate as illumina reads. Calling SNPs on such assemblies would result in lots of false positives. The authors mention SNPs belonging to dense clusters are considered false positives, but it was not clear to me what that exactly means and how that would capture inaccuracies due to base level errors.

- The manuscript does not discuss the coverage requirements per strain to run Strainberry. For example, at what sequencing coverage, the difference between the strains is no more identifiable and how this relates to the number of strains in the sample. I would like to see such analysis on one of the mock communities used for evaluation by 1) downsampling the data to lower depth of coverage and 2) by changing the number of strains.

- The authors provide a nice evaluation framework for strain resolved assemblies. I would highly recommend the authors to provide that as a standalone tool. As mentioned in the manuscript, there are very few ways to evaluate such assemblies and providing such a tool would be a good contribution.

Automated strain separation in low-complexity metagenomes using long reads

RESPONSES TO REFEREES

We would like to thank the reviewers for their constructive remarks and suggestions which led us to improve the manuscript. This document is a point-by-point response to all comments on our initial manuscript (reviewers' comments in red, our replies in black). We briefly summarize all major changes done to the manuscript in the following list:

- We implemented Strainberry as an iterative algorithm that no longer requires the user to specify a number of strains. We observed that results either remained very similar or improved. For this reason, we updated all the results. Nevertheless, the main conclusions do not change, as the new version of Strainberry continues to reliably separate strains.
- Due to the aforementioned modification in the Strainberry's pipeline, we updated the "*Haplotype phasing and read separation*" section to no longer use the Freebayes and WhatsHap-polyphase tools, which one of the reviewers had difficulties compiling. Strainberry now only relies on Longshot and HapCUT2 for calling and phasing SNVs. We introduced the "*Iterative separation*" section in the "*Components of the strain separation pipeline*" part of the Methods. The Discussion was also updated, removing the paragraph mentioning the prior knowledge of the strain multiplicity as a limitation of the tool.
- A new Results section "*Influence of strain coverage, divergence, number of strains, and recombination rate on strain separation*" was created to present all the evaluations suggested by the reviewers. A paragraph in the "*Datasets and preparation*" section of the Methods was introduced in order to describe how the datasets were generated. Figure 4 was added to highlight some key aspects of the results (related assembly evaluation metrics were added to Supplementary File 1). Supplementary Table 2 was finally introduced to provide accession codes of the reference sequences used in these latest analyses.
- Figure 5 of our initial submission has been moved to Supplementary Figure 4.

REVIEWER COMMENTS

Reviewer #1 (Remarks to the Author):

The authors present a software pipeline for the assembly of genome sequences from metagenomic sequences, which represent mixtures of closely related strains. As many microbial species occur as multiple strains in their natural environments, strain separation is a major problem in metagenomic assembly and binning. The manuscript discusses existing, mainly short-read techniques and introduces a workflow that is based on long-read assembly and haplotyping. The approach is limited to low complexity metagenomes and the software requires previous knowledge about the number of strains. Therefore, Strainberry seems to be an intermediate step on the long way towards fully strain-resolved metagenomics.

The manuscript provides very interesting data, which justify most of the conclusions. However, it leaves a couple of major questions open:

1. How different can strains be, to be assembled by Strainberry? The mock community experiments don't answer this question sufficiently.

The ability to assemble strains depends on the divergence of the two strains and the ability of the phasing algorithm to "link" SNVs. By default, we separate reads only in phased regions having at least the 0.1% of called positions which is a first lower bound on the ability to assemble strains. In order to improve the analysis of Strainberry and provide an answer to the question, we created 7 small artificial mock communities with simulated reads. These communities contain a ~50X coverage of the Escherichia coli K-12 strain and a ~50X coverage of a different Escherichia coli strain. The latter having increasing degrees of divergence with respect to strain K-12. More precisely, we considered known reference genomes having the following divergences: 0.02%, 0.2%, 0.3%, 0.4%, 0.5%, 0.8%, and 1.0%.

Each artificial community was assembled with Flye and separated with Strainberry. Results are presented in the Section "Effects of strain coverage, divergence, number of strains, and recombination rate on strain separation". More precisely, the following paragraph has been added:

"Strainberry's separation of low-divergence strains was evaluated on seven two-strain communities consisting of the E. coli strain K-12 and a second E. coli strain characterized by an increasing divergence. Datasets were generated with PacBio-like reads that were simulated from known reference genomes (see Supplementary Table 2). In this case, Strainberry fails to separate strains with low strain divergence (likely due to its default behavior of not separating regions with a low percentage of SNVs). A significant improvement of the reference coverage is observed at 0.39% of strain divergence, while a near-complete assembly is reached at 0.50% of divergence. In all these datasets, the average nucleotide identity of separated sequences is always greater than 99.8% and a significant improvement compared to the strain-oblivious Flye assembly (Figure 4)."

Figure 4 compares Strainberry and Flye assemblies in terms of average reference coverage and average nucleotide identity.

2. How rare can strains be, to be assembled by Strainberry? Many metagenomes consist of one or few abundant strains and many rare strains.

We generated 12 mock communities which are downsampled versions of the Mock3 dataset (*B. cereus*, *E. coli* strain K-12, and *E. coli* strain W). More precisely, the downsampling was performed according to the two following strategies:

- uniform (same depth of coverage for each strain)
- uneven (only *E. coli* strain W was downsampled)

For both downsampling strategies, we considered six increasing depths of coverage (5X, 10X, 20X, 30X, 40X, and 50X). When the only *E. coli* strain W was downsampled, the other two bacterial strains had a fixed 50X depth of coverage.

Each downsampled community was assembled with metaFlye and separated with Strainberry, as done for the other mock communities. Results are presented in the Section “Effects of strain coverage, divergence, number of strains, and recombination rate on strain separation”. More precisely, the following paragraph has been added:

*“The influence of strain coverage was tested on mock communities generated by downsampling the Mock3 dataset at 5X, 10X, 20X, 30X, 40X, and 50X. Two downsampling approaches were considered: a uniform depth of coverage of the three strains, and an uneven depth of coverage. The latter consists in applying the aforementioned downsampling levels to *E. coli* strain W, while keeping the other two strains at 50X. Figure 4 shows, for both the uniformly and unevenly downsampled datasets, that the quality of Strainberry separation reaches a plateau after 30X coverage and is mediocre at 10X coverage. A near-complete reference representation (>95%) and a high sequence identity (>99.8%) thus requires at least 20X coverage. Also, uneven coverage of conspecific strains does not seem to have an impact on Strainberry’s ability to assemble their genomes which is only affected by absolute depth of coverage.”*

Figure 4 graphically compares Strainberry and Flye assemblies in terms of average reference coverage and average nucleotide identity.

The aforementioned experiments also address remark 4.1 from Reviewer #2.

3. How much does the assembly result depend on the mode of evolution between strains (e.g., amount of recombination compared to inversion and translocation)?

This is a good point that we had not looked at. In addition we now evaluate Strainberry on a simple simulated mock community based on reference genomes of species characterized by different recombination rates. We added the Section “Effects of strain coverage, divergence, number of strains, and recombination rate on strain separation” and the following paragraph in the Results:

“We also performed an experiment to estimate the influence of historical recombination on strain separation, not including recent recombination events among strains in the same community as they fall outside of our focus on low-complexity communities. Strainberry’s ability to discriminate strains depends on the average frequency of SNVs. With higher recombination rates the SNV distribution becomes overdispersed. Nevertheless, recombination tracts tend to be small (a few Kbp, often less) and are expected to be mostly spanned by long reads. In order to evaluate the performance of Strainberry on strains characterized by different historical recombination rates²⁶, we simulated a mock community based on the complete reference genomes of two Buchnera aphidicola strains (low recombination rate), two Escherichia coli strains (average recombination rate), two Helicobacter pylori strains, and two Neisseria meningitidis strains (high recombination rate). As opposed to Flye, which provided strain-oblivious assemblies for the species characterized by average and high recombination rates, Strainberry was always able to separate strain genomes. Nevertheless the average nucleotide identity of separated assemblies slightly decreases with higher recombination rates (Supplementary File 1).”

Also, inversions and translocations are handled naturally by our method. The phasing algorithm (HapCUT2) creates blocks. Inversions and translocations induce breaks between blocks and our assembly separation is applied to each haplotype within each block.

4. What criteria apply to long-read sequencing data to be used by Strainberry? Many datasets are of poor quality - can they be used or are there minimal criteria e.g. for coverage and sequence fractions that can be aligned to the initial assembly? Or should the user just try?

Coverage is the main factor in the ability to accurately assemble strains. If a strain has a low abundance (e.g., a depth of coverage less than 10X), any assembler would struggle to completely and accurately reconstruct its genome due to the lack of coverage and, at the same time, to the high error rate of long-read sequencing. Thus, we have added the following sentences in the Discussion:

“Strainberry can be applied to any long-read dataset of sufficient quality, i.e. for which an initial strain-oblivious assembly can be obtained. In our tests, any strain present at coverage above 10X ends up being correctly separated. The quality of the resulting Strainberry strain separation will also depend on the quality of the initial strain-oblivious assembly.”

Minor issue:

1. Supplementary Figure 3 lacks a unit for SNP density.

Thank you for spotting this. We modified the x- and y-axis labels as follows:

- contig length (Mbp)
- SNV density (SNVs per 100 bp)

Technical issue:

We failed to install strainberry. All prerequisites were installed.

In the environment.yml, the line that doesn't work is the following:

- pip:
- git+<https://github.com/whatshap/whatshap.git>@polyploid-haplotag

When we replace that line with:

-whatshap

It does install as I mentioned, but then the multi-strain separation doesn't work.

The error during the installation is the following:

ERROR: Failed building wheel for whatshap

ERROR: Could not build wheels for whatshap which use PEP 517 and cannot be installed directly

Here is the part of the error message which makes me think that the problem is in the code and not in our system or the installation procedure:

In file included from src/polyphase/trianglesparsematrix.cpp:1:

src/polyphase/trianglesparsematrix.h:15:5: error: 'uint64_t' does not name a type
15 | uint64_t entryToIndex(uint32_t i, uint32_t j);
| ^~~~~~

| ^~~~~~

src/polyphase/trianglesparsematrix.h:16:5: error: 'uint64_t' does not name a type
16 | uint64_t size();
| ^~~~~~

| ^~~~~~

src/polyphase/trianglesparsematrix.h:17:15: error: 'uint32_t' has not been declared
17 | float get(uint32_t i, uint32_t j);
| ^~~~~~

| ^~~~~~

src/polyphase/trianglesparsematrix.h:17:27: error: 'uint32_t' has not been declared
17 | float get(uint32_t i, uint32_t j);

And there are many more instances of similar errors.

We thank the reviewer for spotting this bug in a dependency. Indeed replacing the line in environments.yml to just 'whatshap' is expected to cause problems when running Strainberry. This happens due to the fact that the latest WhatsHap version does not yet integrate the polyploid feature of the 'haplotag' command.

After a careful look at the installation errors mentioned in this remark, it appears the problem was not from Strainberry code but, instead, it is related to a software dependency. We believe this technical issue could be solved by making sure the installed C++ compiler is fully compatible with the C++11 standard (e.g. GCC version >= 4.8.1).

Nevertheless, in order to address the remark #2 from Reviewer #2, we have improved Strainberry and the latest version (1.1) no longer requires the WhatsHap dependency, and, as a byproduct, the creation of Strainberry's conda environment is no longer expected to produce the errors mentioned by the reviewer. Moreover, we defined a Continuous Integration procedure on the Github repository which verifies that Strainberry could be successfully installed and run with the example data.

Reviewer #2 (Remarks to the Author):

In this paper, the authors propose a new method called Strainberry that uses long reads to identify strains from metagenomic samples and evaluate the accuracy of Strainberry on both synthetic and real metagenomic datasets. Strain resolution is a very important problem in metagenomics and the authors provide good insights into how this problem can be solved using emerging use of long read technologies in metagenomics. The manuscript is very well written and easy to follow. The code for the method is provided on GitHub with detailed instructions to install and run. Overall, the manuscript contributes significantly to the problem of strain resolution.

Here're some of the comments that need to be addressed before I can recommend the manuscript for publication.

1. Strainberry method is very much similar to one of the recent methods published for the haplotype assembly of Human genome - <https://www.nature.com/articles/s41587-020-0711-0> . I would recommend the authors to cite this paper and mention that similar approaches have been used for single genome assembly.

This is a good point. We have added the following sentence and citation to the aforementioned study (as well as another one from Porubski *et al.*) in the discussion:

"Notably, two recent pipelines that also rely on those techniques produced chromosome-scale haplotype-resolved single human assemblies^{32,33} using additional sequencing data (Hi-C or Strand-seq)."

We note also that key methodological ingredients in those two recent single-human chromosome-scale assemblies are the integration of additional long-range data such as Hi-C and Strand-seq data, which are outside the scope of Strainberry.

2. Strainberry requires the number of estimated strains in the sample as an input from the user. Most of the evaluation presented in the paper was based on the samples with the known number of strains for a particular species. However, this seldom happens in real metagenomic samples. the authors allude to this point in the discussion, but some evaluation is required on this. For example, if one runs Strainberry with different values for the number of strains, how they can determine what's the right number of strains.

A simple way to determine the right number of strains could be to look at the average distance between the mapped reads and the phased haplotypes. In order to address this

remark and improve the tool usability, we implemented in Strainberry an automated procedure which is able to produce comparable or significantly better assemblies without the need to specify the number of strains in input.

More precisely, Strainberry's pipeline has been updated by using Longshot+HapCUT2 in an iterative way and using the average Hamming ratio (i.e., normalized Hamming distance) between reads and their closest haplotype. If the average Hamming ratio does not improve significantly after an iteration, the assembly obtained in the previous iteration is returned in output. Moreover, at each iteration, only contigs for which such a metric significantly improves are separated.

The "Components of the strain separation pipeline" section of the Methods section was updated to take into account this new implementation. In more detail, the subsection "Haplotype phasing and read separation" has been significantly modified and the subsection "Iterative separation" was introduced to explain the iterative process.

This newer implementation produced either similar results or improved the strain-separated assembly when more conspecific strains were present. When only two conspecific strains were present, Strainberry always stopped after the first iteration. For more complex datasets, Strainberry improved the strain-separated assembly by performing a sufficient number of iterations. As an example, for the Mock9 dataset, the new iterative version of Strainberry was able to produce a high quality reconstruction of the two *S. aureus* strains while also improving the reference coverage for the other three similar strains (*S. sonnei* and the two *E. coli* strains). Finally, applying Strainberry independently on the 11 high-quality bins of the HSM dataset yielded either a very similar result or a larger separated-assembly.

3. The authors use bacterial genome assemblies generated using long reads as a starting point for Strainberry. However, these assemblies tend to have low base level accuracy as the reads are not as accurate as illumina reads. Calling SNPs on such assemblies would result in lots of false positives. The authors mention SNPs belonging to dense clusters are considered false positives, but it was not clear to me what that exactly means and how that would capture inaccuracies due to base level errors.

According to the Longshot manuscript, dense clusters of SNVs could appear due to systematically mismapped reads. We also noticed that retaining SNV calls with low QUAL value (i.e., less than 50) or SNVs that were tagged by Longshot as belonging to a dense cluster often caused a failure in Whatshap-Polyphase execution. As the improved version of Strainberry (developed to address the limitation of the prior strain multiplicity knowledge) no longer uses Whatshap-Polyphase, this filter is no longer applied. The corresponding sentences in the "Haplotype phasing and read separation" section of the "Methods" were thus removed.

4. The manuscript does not discuss the coverage requirements per strain to run Strainberry. For example, at what sequencing coverage, the difference between the strains is no more identifiable and how this relates to the number of strains in the

sample. I would like to see such analysis on one of the mock communities used for evaluation by 1) downsampling the data to lower depth of coverage and 2) by changing the number of strains.

We have addressed point 1) as part of remark #2 from Reviewer #1. In a nutshell, we have evaluated Strainberry's ability to separate strains on Mock3 by lowering the depth of coverage of all strains (*B. cereus*, *E. coli* strain K-12, and *E. coli* strain W) and by lowering only the depth of coverage of *E. coli* strain W.

For point 2), we created four simulated mock communities based on 2, 3, 4, and 5 *E. coli* strains having roughly the same levels of pairwise divergence (0.7%-1.4%). Each simulated dataset was assembled with Flye and separated with Strainberry. Our results indicate that Strainberry is able to handle the separation of up to 5 strains albeit with some noticeable loss of quality when separating 5 strains compared to the separation of 3 strains: the average reference coverage drops from 95% (3 strains) to 75% (5 strains), while the average nucleotide identity of sequences with respect to their closest reference drops from 99.9% (3 strains) to 99.6% (5 strains).

Results are presented in the Section "Effects of strain coverage, divergence, number of strains, and recombination rate on strain separation". More precisely, the following paragraph has been introduced:

"In order to evaluate Strainberry with respect to the number of conspecific strains, we created four simulated mock communities consisting of 2, 3, 4, and 5 E. coli strains characterized by pairwise divergences ranging from 0.7% to 1.4%. Each simulated dataset was assembled with Flye and separated with Strainberry. Figure 4 indicates that Strainberry is able to handle the separation of up to 5 strains albeit with some noticeable loss of quality when separating 5 strains compared to the separation of 3 strains: the average reference coverage drops from 95% (3 strains) to 75% (5 strains), while the average nucleotide identity of sequences with respect to their closest reference drops from 99.9% (3 strains) to 99.6% (5 strains). Nevertheless in all these scenarios Strainberry improved upon the Flye assembly (Supplementary File 1)."

Figure 4 graphically compares Strainberry and Flye assemblies in terms of average reference coverage and average nucleotide identity.

5. The authors provide a nice evaluation framework for strain resolved assemblies. I would highly recommend the authors to provide that as a standalone tool. As mentioned in the manuscript, there are very few ways to evaluate such assemblies and providing such a tool would be a good contribution.

This is a good suggestion, however we feel that it deserves a longer treatment than an addition to this article. In fact, we had started a discussion with the CAMI metagenome challenge organizers, the MetaQUAST team, and the metaFlye lead developer on what would be a reasonable way to evaluate metagenome assemblies in the presence of strains. The discussion led to a draft online document, which we link here just for information:

<https://docs.google.com/document/d/1pHWuQFWxLIk0g0qEKnc6rGeTBa6H87nm-LmSJtvH4A/edit>

As this effort is maturing, we will consider either releasing a stand-alone tool or use an improved version of MetaQUAST. In the Strawberry manuscript, our evaluation framework is open-source yet unfortunately tailored to the specificity of each dataset. While we do highlight some potentially useful directions for strain evaluation, we do not yet have a sufficiently original nor general standalone evaluation tool.

REVIEWERS' COMMENTS

Reviewer #1 (Remarks to the Author):

The authors have carefully addressed all issues, which is very much appreciated.

Reviewer #2 (Remarks to the Author):

I am happy with the revisions and recommended the manuscript for publication.

RESPONSES TO REFEREES

No other issue was raised by the referees on the revised manuscript. We thank them again for their constructive remarks and suggestions.

REVIEWERS' COMMENTS

Reviewer #1 (Remarks to the Author):

The authors have carefully addressed all issues, which is very much appreciated.

Reviewer #2 (Remarks to the Author):

I am happy with the revisions and recommended the manuscript for publication.